# Expectations of Patients Recovering from SARS-CoV-2 towards New Forms of Pulmonary Rehabilitation

**DOI:** 10.3390/ijerph20010104

**Published:** 2022-12-21

**Authors:** Mariusz Migała, Bożena Płonka-Syroka, Krystyna Rasławska, Beata Skolik, Izabela Spielvogel, Katarzyna Piechota, Daria Hołodnik, Magdalena Hagner-Derengowska

**Affiliations:** 1Faculty of Physical Education and Physiotherapy, Opole University of Technology, 45-758 Opole, Poland; 2Specialist Hospital of the Ministry of Interior and Administration in Głuchołazy, 48-340 Głuchołazy, Poland; 3Department of Humanities and Social Sciences, Department of Pharmaceutical Humanities, Medical University of Piastów Śląskich in Wrocław, 50-367 Wrocław, Poland; 4Faculty of Medical Sciences, The University of Applied Sciences in Nysa, 48-300 Nysa, Poland; 5Department of Physical Cultore, University of Nicolaus Copernicus in Torun, 87-100 Torun, Poland

**Keywords:** COVID-19, inpatient rehabilitation, disease experience, disability, convalescence

## Abstract

The purpose of this study was to explore the experiences of patients attending an innovative technology-enhanced pulmonary rehabilitation program of National Health Found Program in Poland. The study included two groups of patients participating in post-COVID-19 stationary rehabilitation. Patients from group I (127 individuals) contracted COVID-19 in 2020, while patients from group II fell ill in 2021 (68 individuals). The study used a self-administered questionnaire. This study was designed as an acceptability study. In the experience related to COVID-19 in both groups of the respondents, the possibility of undertaking inpatient rehabilitation in a hospital ward played an important and positive role. Patients who experienced COVID-19 symptomatically expected that rehabilitation would eliminate the related dysfunctions, such as reduced respiratory efficiency of the lungs, disorders of the nervous system, and cognitive disorders (the so-called brain fog). All respondents who experienced symptomatic COVID-19 positively assessed the rehabilitation program offered. Among the highest-rated rehabilitation, elements were identified: exercise on a cycle ergometer implemented with video stimulation, group fitness exercises, and breathing exercises. Other innovative forms of rehabilitation were positively evaluated by 10% to 25% of patients.

## 1. Introduction

The research on how patients experience the COVID-19 epidemic is an important addition to clinical study on the disease [1]. The aim is to gain insight into the patients’ perspective on how they define the risk, and then implement individual protection strategies. The analysis of these issues may become a basis for the formulation of effective epidemic prophylaxis programs and state epidemic management strategies based on the clinical standard (Evidence-Based Medicine). The developers of these programs and strategies should take into account the state of health awareness of the patients and adapt the recommendations to the possibility of their assimilation by laymen in order to conduct accordingly. The possibility of examining the health awareness of laymen creates a modern standard of medical sociology, oriented, inter alia, to the analysis of the experience of the disease by the patients [2,3]. It includes the fact that, for the patient, the disease is not only a somatic experience, but, above all, an existential one significantly changing the way he or she functions in a society. It is also a source of suffering from which the patient wants to protect themself by creating individual defense strategies to a greater or lesser extent consistent with the clinical standard. In the case of communicable diseases, this type of compliance is essential to the health safety of the population; therefore, care should be taken to ensure that epidemic threat content is communicated to the public in a timely and understandable manner. Members of the population defining the epidemic risk in clinical terms follow the prophylactic recommendations of specialists, do not avoid undergoing treatment, and recover more easily after the disease. Those who do not understand how to define an epidemic risk in clinical terms negate the risk itself or create ways of defining it that are not in line with the clinical standard. As a result, they do not implement the methods of prophylaxis recommended by specialists, avoid undergoing clinical treatment, and, when symptoms of an infectious disease occur, they fight them with self-treatment methods. With severe symptoms of the disease, patients feel a strong fear; hence, the disease that they reject and do not understand becomes a serious disturbance in their life. By failing to comply with anti-epidemic recommendations, they also pose a threat to other members of the population.

The purpose of this study was to explore the experiences of patients attending an innovative technology-enhanced pulmonary rehabilitation program of the National Health Found Program in Poland. The ways of experiencing the disease by patients before and after the diagnosis, patients’ relations with the medical personnel providing care for them, expectations towards rehabilitation, and the assessment of the benefits obtained were analyzed.

## 2. Materials and Methods

### 2.1. Participants

The study covered two groups of patients, staying on post-COVID-19 rehabilitation at the Specialist Hospital of the Ministry of Interior and Administration in Głuchołazy named after St John Paul II in the years 2021 and 2022. This facility, since September 2020, as the only one in the country, has been implementing a pilot program in the field of therapeutic rehabilitation for beneficiaries with a history of COVID-19 disease. Patients were retrospectively divided into two groups: group I fell ill with COVID-19 in 2020 before the introduction of vaccinations or did not have time to be vaccinated at the beginning of 2021, while patients from group II fell ill in 2021 after the introduction of preventive vaccinations in Poland. They stayed in rehabilitation for up to 12 months after suffering the disease. Groups I and II included 127 and 68 people, respectively. The study included a group of approximately 20% of patients undergoing rehabilitation during the analyzed period. Patient participation in the study was anonymous and voluntary. The research was approved by the Bioethics Committee at the Medical University of Piastów Śląskich in Wrocław (research project entitled Analysis of the experiences of patients undergoing rehabilitation related to the COVID-19 disease, registration number of the Center for the Support of Science BW-69/2020) and the Director of the Specialist Hospital of the Ministry of Interior and Administration in Głuchołazy. The patients gave written consent to fill in the questionnaires. Expressing their feelings and opinions in unstructured interviews, they also maintained anonymity and the voluntary nature of contact with the researcher.

### 2.2. Methods

The research was conducted in the form of questionnaires. The questionnaire was not validated. Due to the dynamic situation related to the pandemic, it was prepared for the purposes of research, based on previously conducted research and experience. The available questionnaires were also analyzed; however, it was decided to use the authors’ one to cover many aspects related to rehabilitation stay in hospital. The premise of the study was not only statistical analysis, but also consideration of opinions expressed by patients during the rehabilitation period (quality actions).

### 2.3. Rehabilitation Program

Patients participated in a short-term, high-intensity stationary pulmonary rehabilitation program based on the COPD patient program [4]. Components were performed once a day and lasted 15–30 min, five times a week for three weeks. The pulmonary rehabilitation program has been described in detail in previous studies [5]. Both groups participated in a pulmonary rehabilitation program enhanced by innovative technologies.

#### 2.3.1. Conventional Pulmonary Rehabilitation

The traditional form of exercises included in the rehabilitation program covered: general co-ordination exercises with a physiotherapist, breathing exercises, and diaphragm exercises with resistance; effective coughing exercises and prolonged exhalation exercises; chest percussion; and inhalation with a 3% NaCl isotonic solution, likewise, Nordic walking.

#### 2.3.2. New Forms of Additions to Pulmonary Rehabilitation

Innovative elements of the program included activities employing virtual reality. Three exercise groups were included for this purpose: capacity training, relaxation training, and cognitive function training. The exercises on a stationary cycle ergometer were incorporated with a projected video simulating a bicycle ride on a 75″ TV to achieve a training heart rate consistent with individual selection during qualification for rehabilitation (Figure 1). The relaxation in VR was performed with VR TierOne device. This system has previously been used in the course of pulmonary [6] as well as cardiac rehabilitation [7]. The cognitive function training in VR involved short-term memory exercises, and eye–hand co-ordination and general fitness training in virtual reality. Both a non-immersive system and immersive using HMD goggles were utilized. To the best of our knowledge, such an extensive program has not been used before in post-COVID-19 patients.

### 2.4. Data Analysis

In the analysis of the presentation of the COVID-19 epidemic, the critical and comparative method was used. The qualitative analysis concerned the respondents’ answers to specific questions and the opinions they added to them. The authors analyzed these statements to find out whether and how the picture of the epidemic among the respondents was influenced by the standard of clinical medicine and what factors influenced it. Quantitative and qualitative methods were used in the analyses of the statements, which seem to be particularly useful in the studies of patients’ experiences related to the disease [8], in studies of the level of patient satisfaction with the obtained medical help and in their evaluation of the quality of treatment and rehabilitation services [9]. In contemporary medical sociology, the analysis of the patient’s experience related to the disease is considered an important element in the diagnostic process [10]. It can also be useful in the course of therapy. The physician’s ability to learn about the patient’s perspective related to disease recovery, mainly in the case of chronic diseases, can also be useful in the treatment of infectious diseases with serious course and consequences requiring rehabilitation, including COVID-19 [11]. This justifies the purposefulness of undertaking multidisciplinary research on the experiences of the patients. Within the standard of the so-called narrative medicine, the possibility of the patient’s insight into the essence of their own disease is considered prognostically favorable [12]. Such insight allows the patient to understand and accept the real parameters of their clinical condition, which increases compliance with their preventive and therapeutic recommendations based on trust in the doctor. In this way, it is possible to limit the occurrence of the no-compliance effect and reduce the general sense of anxiety in patients. The method allowing the patient to gain an insight into the essence of the disease creates an individual narrative about the disease, which gives the possibility of shaping the experience related to the disease in the form of a structured story [13]. Presentation of these experiences by the patients is possible during formal psychotherapy, in informally created groups of patients, and in a written form [14].

## 3. Results

### 3.1. Characteristics of the Patients (Group I)

The first group of respondents consisted of patients who contracted COVID-19 in 2020. The group included 127 people (67 women and 60 men). It was dominated by respondents who were currently married (38.6% married men, 33.2% married women) or were in the past (11% widowers and widows, 18.1% single persons). The largest group were patients aged 50–59 (40.9%), then 60–69 (26.8%), 40–49 (13.4%), and 70–79 (11.8%), 30–39 years of age (5.5%), and 20–29 years (1.6%). Therefore, the majority of respondents were mature people (aged 40+ 92.9%), staying 10 years or more in the labor market, with a stable professional position. Families with two children (44.9%) dominated, followed by one child (26.8%), three (14.2%), four (0.8%), and five children (1.6%), while 11,8% of the respondents were childless. In the first group of respondents, 3.1% had higher education, and 9.4% had completed post-secondary one, 68% had secondary school-leaving examination, and 15% had vocational qualifications. Most of the respondents from group I lived in cities: 22.8% from 5000 to 50,000, 20.5% from 100,000 to 500,000, 10.2% above 500,000, and 5.5% below 5000.

### 3.2. Characteristics of the Subjects (Group II)

The second group of respondents consisted of 68 people (34 women and 34 men). It was dominated by married people, currently (35.29% married men, 26.47% married women) and married in the past (10.29% widows, 2.94% widowers). Single people of different status constituted a smaller group (1.47% unmarried, 2.94% single men, 7.35% women and men describing themselves as “free”, 1.47% of the respondents declared themselves as divorced, and the status of 1.47% was unspecified. Families with two children (48.53%) dominated, followed by one (22.06%), three (13.24%) and four (1.47), while 10.29% of the respondents had no children. In group II, the education of the respondents was even higher. In total, 35.29% had graduated from higher education institutions, 7.35% post-secondary schools, 35.29% secondary schools with high school diploma, and 19.12% vocational schools. In group II, the housing conditions were considered to be very good too. A total of 55.9% lived in an owner-occupied flat, 23.5% in their own house with an area of more than 100 m^2^, and 7.4% in their own house up to 100 m^2^. Only 10.3% lived in accommodation flats.

In 98% of respondents from group I, COVID-19 disease was symptomatic, while, in group II, the percentage was 91.18%. Therefore, the respondents did not experience the negation of the existence of COVID-19 disease, the risks associated with it, and the need to take any protective measures, which is characteristic of many members of the Polish population. Table 1 presents the health status of the subjects starting rehabilitation.

In the experience related to COVID-19 in both groups of the respondents, the possibility of undertaking inpatient rehabilitation in a hospital ward played an important and positive role. Patients who experienced COVID-19 symptomatically expected that rehabilitation would eliminate the related dysfunctions, such as reduced respiratory efficiency of the lungs, disorders of the nervous system, and cognitive disorders (the so-called brain fog). The patients also expected an overall improvement in well-being, an improvement in physical condition, a return to the full physical condition, and receiving instructions on how to do so. They expected an improvement in the parameters related to mental and social functioning, that is, reducing or eliminating the brain fog, stress, and anxiety. They also wanted to learn more about recovering from the COVID-19 disease and preserving health. Patients’ expectations regarding inpatient rehabilitation should be considered realistic and adjusted to their knowledge of the nature of COVID-19 and the complications associated with this disease. Patients were aware of the biological causes of the disease, as well as its effects, which they also defined in clinical terms. They expressed their readiness to actively participate in the proposed exercise program and believed that their individual rehabilitation would prove effective. A similar attitude was found in the second group of respondents (Table 2).

Respondents were asked to evaluate the rehabilitation methods used at the facility and they could choose several answers. The results are presented in Figure 2. 

The possibility of choosing unconventional forms of rehabilitation was viewed positively by 47.06% of respondents, while 35% did not view it positively. Among the methods of psychological rehabilitation, respondents were most likely to receive individual psychotherapy (35.29%), art therapy—art therapy (23.53%), and group psychotherapy (17.65%) during their hospital stay (Figure 3).

Some of the patients undergoing rehabilitation in the hospital regime perceived the proposed model negatively. They considered the expectations related to the possibility of free walks, establishing contacts with other patients, and getting closer to them on a social basis as insufficiently satisfied. There were no special spaces for this purpose in the hospital. The expectations of this group came close to the spa treatment model, in which the above-mentioned needs are successfully met in specially designated facilities and places. Some patients who underwent COVID-19 asymptomatically and did not develop serious physical dysfunctions critically referred to the rehabilitation model proposed in Głuchołazy for other reasons. They did not feel any decline in physical condition or any measurable ailments. These patients were not interested in doing many exercises each day as part of the individual rehabilitation program that was offered to them. This applied to both exercises with the use of modern equipment and typical gymnastic activities. Patients treated them with reluctance because they did not feel any limitation in their physical fitness, so they did not see the need for physical improvement (Table 2).

Furthermore, respondents were asked about the additional elements of the program that could be expanded. The answers are presented in Figure 4.

## 4. Discussion

Constructing the picture of COVID-19 disease in the minds of patients was of a social nature. For most of the respondents, the very existence of the disease was unmistakable. They perceive the risks associated with the disease in a realistic manner, which directs them towards constructive resolution of health and social problems. These fears reflect the fundamental threads present in 2020 in the Polish press and media discourse, which the respondents internalized and they became a source of fears and suffering for them [1]. Patients’ experiences related to COVID-19 were significantly influenced by the manner of reporting the epidemic in the Polish media, which is discussed in more detail in the monograph devoted to it. An appropriate standard of information about the epidemic was neither developed nor presented to Poles in an unambiguous and understandable way. As a result, the respondents from both surveyed groups were not aware that the epidemic could threaten them and, when they fell ill with COVID-19, they did not have an effective management strategy. The respondents from both groups, due to the high level of education, expressed a higher-than-average level of acceptance for presenting the epidemics in clinical terms. They were independently seeking information about COVID-19 from various sources, including 15% from foreign television. In January 2020, the Internet was the main source of the news for only 3.1% of them. When the epidemic broke out in China, only 65.4% of respondents considered the information about COVID-19 to be alarming. When the epidemic moved to Europe in February 2020, 88.9% were worried about this fact, and, at the end of February, this was already 92.1% of respondents. The official information strategy on COVID-19, which was present in the Polish media from January to March, was assessed negatively by the respondents. They stated that they had not been adequately prepared by the country’s authorities for the outbreak of the epidemic in Poland. As many as 24.4% were completely surprised by the outbreak of the epidemic in Poland and 9.4% by its dynamic development in our country. The government media dominated by the “official optimism” was blamed for the Poles’ unpreparedness for the epidemic. Therefore, the respondents still searched for information from various sources, including 48% from the Internet. However, only 9.44% of respondents drew information about COVID-19 only from this source. This explains the maintenance of a high level of trust in the standard of clinical medicine among the respondents from group I and the lack of popularity of alternative treatment and prophylaxis methods among them.

The results obtained in the research on the disease experiences by patients suffering from COVID-19 in the first year of the epidemic in Poland are compatible with the state of the art. It is stated in the literature that properly planned and implemented social policy enables patients to fully concentrate on matters related to the disease and procedures enabling recovery or rehabilitation without fear of losing their social position before the onset of the disease or means of subsistence, as well as ensuring the existence of the family. In the conditions of social security provided by the law, during the period of illness, the patients rehabilitated in Głuchołazy could properly respond to the diagnosis made by the doctor and accept the individual rehabilitation program offered to them. There was no conflict among the respondents between trusting the doctor and the necessity to reject their recommendations (or even diagnosis) resulting from the lack of social security during the period of illness and rehabilitation. Patients’ health awareness could be formed in the circle of rational views expressed by specialists and be based on the modern scientific standard, so it had a chance to become effective [15].

The methods of clinical rehabilitation offered to patients after COVID-19 were based on scientific grounds. An individual rehabilitation program enhanced with new technologies carried out in the hospital setting corresponded to the characteristics of such therapy and activated various areas of the patient’s identity, not only their physical fitness, but also the area of aesthetic and emotional experiences (music therapy and exercises integrated with the use of virtual reality) [16]. Moreover, VR-based methods also appear to lead to lower stress levels during exercise [17], as well as effectively acting as a distractor during stressful patient situations [18,19]. The effectiveness of the methods of clinical rehabilitation offered to the patients was, for many of them, a sufficient barrier against discouragement related to the persistence of post-COVID-19 dysfunctions and the search for methods to overcome in the area of the so-called alternative medicine and self-medication that allowed the related health risks to be avoided [20].

For many patients, the availability of religious services in the hospital was also very important, such as contact with the hospital priest (also by phone), the possibility of participating in the Holy Mass, and individual prayers in the hospital chapel. This helped some patients to eliminate certain religious practices that were prognostically unfavorable and misconceptions leading to increased anxiety, guilt, and loneliness [21]. Properly implemented pastoral support offered to the patients is beneficial in the future. It can also lead to a reorientation of a patient’s life after suffering a serious and fatal illness. That period in life does not have to run under the influence of post-traumatic stress disorder, but it can—with properly provided help—take on a new, satisfying form.

The conducted studies also confirm previous analyses of the experiences of chronically ill patients, in which the type and scope of social support provided to them had a significant influence on the way they experienced the disease [22]. When it was supported in a sufficiently wide range and in a timely manner, the patient could develop an effective strategy enabling the internalization of the disease in their individual biography, could learn to live with it, and achieve many possible types of satisfaction. Trauma resulting from the disease may become the basis of post-traumatic growth for many patients undergoing appropriate treatment and rehabilitation. Life after illness, even despite persistent dysfunctions, can be a source of satisfaction and happiness for the patient.

Some limitations of the study should be noted. First, the research method should have been validated but, due to the need to quickly evaluate the program, this step was not adhered to. Second, we recorded that not all patients answered all the questions. Third, using a standardized survey method would improve the quality of the study. Fourth, follow-up after hospital discharge would be both interesting and provide additional clinical information.

## 5. Conclusions

All patients experienced COVID-19 as a somatic and private experience. The respondents negatively assessed the general standard of information about the epidemic provided by the official state institutions. They recognized that it was dominated by “official optimism”, which was not adequate neither to the scale of the threat nor to the real epidemic situation in the country. Observing the improvement of physical capacity in consecutive days and weeks of exercise became a very positive experience for the respondents, as was watching the recovery of fitness in other patients. Among the highest-rated rehabilitation, elements were identified: exercise on a cycle ergometer implemented with video stimulation, group fitness exercises, and breathing exercises. Other innovative forms of rehabilitation were positively evaluated by 10% to 25% of patients. The conclusions from the research can be used to improve the effectiveness of post-COVID-19 rehabilitation and adjust its standard to the patients’ expectations. They can also be useful in the process of creating a state information strategy for the prevention of serious infectious diseases.

## Figures and Tables

**Figure 1 ijerph-20-00104-f001:**
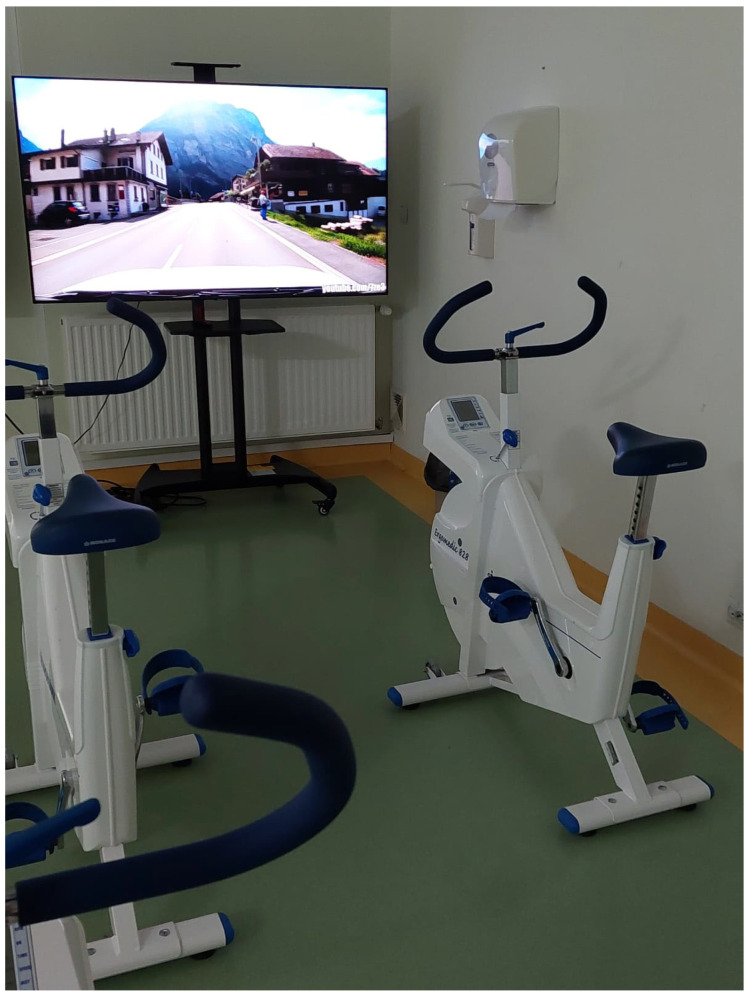
Station for capacity training on a cycle ergometer.

**Figure 2 ijerph-20-00104-f002:**
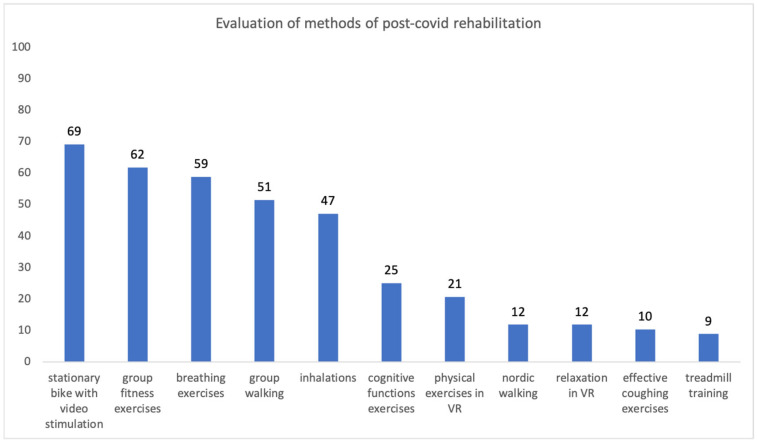
Evaluation of methods of post-COVID rehabilitation.

**Figure 3 ijerph-20-00104-f003:**
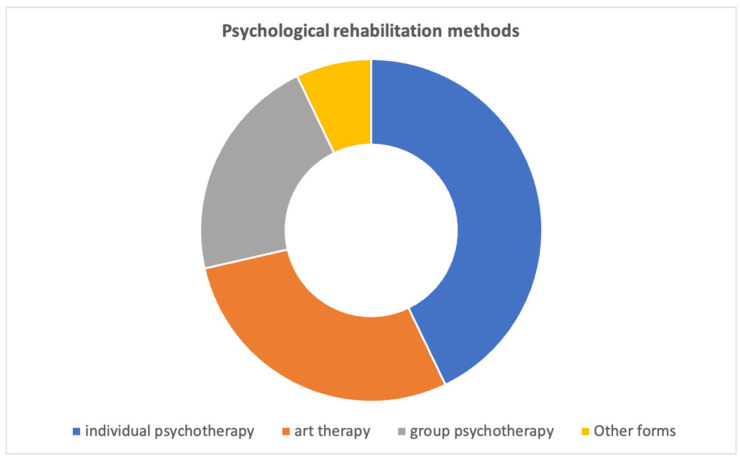
Psychological rehabilitation methods.

**Figure 4 ijerph-20-00104-f004:**
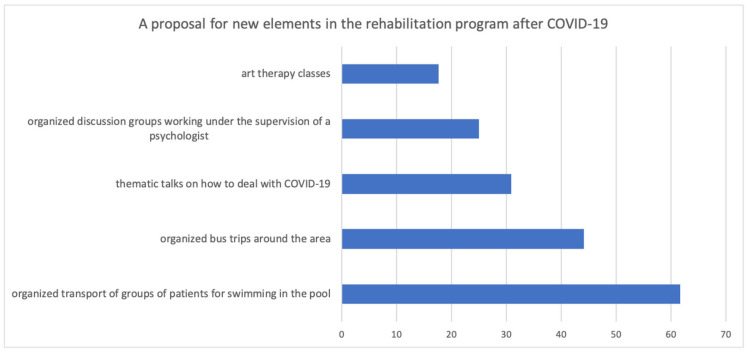
A proposal for new elements in the rehabilitation program after COVID-19.

**Table 1 ijerph-20-00104-t001:** Health status in the subjects starting rehabilitation.

Symptoms	Percentage of Responses
Main symptoms	
difficulty breathing	66.18
physical weakness	60.29
brain fog symptoms (lack of concentration, impaired memory, cognitive functions)	50.0
muscle pain	39.71
headaches and other physical symptoms	22.06
no complaints	8.82
Associated symptoms	
trouble sleeping and waking up several times during the night	58.82
difficulty falling asleep and waking up early in the morning unable to fall back to sleep	30.88
insomnia	19.12
drowsiness	17.65
apathy	11.76
no joy (anhedonia)	11.76
symptoms related to mental state	11.76
increased nervousness	8.82
Nightmares	588

**Table 2 ijerph-20-00104-t002:** Assessment of the provision of other needs during post-COVID-19 rehabilitation.

Need For Help/Care	For (%)	Against (%)
pastoral care	64.71	11.76
medicine	80.88	10.29
Necessity of changes in medical care		
more frequent doctor visits and pulse oximeter examinations of family members. blood pressure and temperature tests	42.65	
thorough examinations during admission and determination of additional treatments	20.59	
No opinion on this subject	17.65	
Satisfaction with physiotherapy care	89.7	2.94
the need to enrich the existing post-COVID rehabilitation program with new, additional elements, which would make it more effective	80.88	2.94
Satisfaction with the conditions of stay		
standard of offered physical exercises	82.35	14.71
meals	80.88	16.18
standard and availability of additional rehabilitation treatments	66.18	32.35
assistance/pastoral care	51.47	45.59
standard of accommodation	50.0	47.06
psychological help	35.29	61.76
A proposal for new elements in the rehabilitation program after COVID-19		
organized transport of groups of patients for swimming in the pool	61.76 44.12 30.88 25.0 17.65	
organized bus trips around the area		
thematic talks on how to deal with COVID-19		
organized discussion groups working under the supervision of a psychologist		
art therapy classes		
Other items		
additional physical exercise after 4:00 p.m.	26.47 25.0 22.06 8.82	
talks on respiratory ways to combat post-COVID-19 ailments		
laryngological and phoniatric classes (rehabilitation).		
do more physical exercise and get some fresh air		

## Data Availability

The data presented in this study are available on request from the corresponding author.

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
