# Peer review of "Expectations of Patients Recovering from SARS-CoV-2 towards New Forms of Pulmonary Rehabilitation"

_ijerph, 2022, doi:10.3390/ijerph20010104_

Round 1

Reviewer 1 Report

Dear authors
Rehabilitation management in Covid-19, particularly in respiratory-related symptoms, is a significant issue that we will be dealing with for many years. Consequently, the evaluation paper addresses the important topic of rehabilitation effectiveness. The work is very interesting but needs some fundamental revisions.
Material and method section
I recommend you to put this section in order and separate the study group's description from the method's description.
results
Please consider presenting some of the described results in tables/graphs etc.
This will facilitate the analysis for the readers.
Similarly, the overly extensive conclusions need to be significantly shortened.

Author Response

Dear Reviewer,

Thank you for your deep and knowledgeable revision of this manuscript. The following are our answers

Reviewer#1, Concern # 1

Material and method section
I recommend you to put this section in order and separate the study group's description from the method's description.

Author response: Thank you for that suggestion. We have changed the division of subsections according to the instructions.

 Reviewer#1, Concern # 2

results
Please consider presenting some of the described results in tables/graphs etc.
This will facilitate the analysis for the readers.

Author response: Thank you for pointing this out, we have included figures.

Reviewer#1, Concern # 3

Similarly, the overly extensive conclusions need to be significantly shortened.

Author response: Thank you for your comment. We agree that the conclusions were too extended, we shortened them.

Reviewer 2 Report

Thank you for the opportunity to review this article. It is very interesting because it examines the acceptability of using new technologies (mainly virtual reality) as support in pulmonary rehabilitation after COVID-19. Its strength is that the study setting was complete, that is, patients were subjected to both conventional rehabilitation and supplemented with new forms of therapy. Such research is necessary to implement new technologies in clinical practice. However, the strength of this study is poorly described. Descriptions of individual interventions should be supplemented. The article also contains minor errors, which are described below.

Detailed review:

1. As it is an acceptability study, it should be stated in the abstract

2. Correct the abstract according to the journal guidelines

3. Move the data analysis section at the end of methods section

4. In tables and results sections, put decimal points instead of commas

5. Table 1. Why were some of the responses divided into for/against and some without division?

6. The first sentence of the conclusion is not necessary. Instead, us 'In conclusion…'

7. Correct the reference according to the journal requirements

8. The main aim of this study was to analyze the determinants of patients attending an innovative technology-enhanced pulmonary rehabilitation program. Because this program is the most important part of the study, it is worth describing it better. Therefore, other researchers/clinicians will be able to adopt it. I am aware that the authors have indicated another publication in which it is described, but it is not fully described there either. I think it would be best to divide it into 'Conventional Pulmonary Rehabilitation', And 'New Forms of Additions to Pulmonary Rehabilitation'. In these subsections, a more precise description of the exercises performed, the cycloergometer video simulator, relaxation training in virtual reality, cognitive training, and eye-hand coordination in virtual reality is provided.

9. If possible, it is worth supplementing the description of the applied interventions with photos of the therapy.

10. In addition, I propose to supplement the data analysis if possible. Perhaps supplement patients’ satisfaction, as well as an intergroup comparison. It seems that the authors have data on satisfaction with individual components of rehabilitation, and perhaps it is worth quoting them in the context of new technologies.

11. All the cited references were from Poland. This is unacceptable. Please refer to the research results and outline the introduction to international literature.

12. The limitations of this study should be discussed.

13. V letter in the title should be written in capital.

Author Response

Dear Reviewer,

Thank you for your deep and knowledgeable revision of this manuscript. The following are our answers

Reviewer#2, Concern # 1

As it is an acceptability study, it should be stated in the abstract

Author response: Thank you for that suggestion. Information has been added.

Reviewer#2, Concern # 2

Correct the abstract according to the journal guidelines

Author response: Thank you for pointing this out. the abstract has been corrected.

Reviewer#2, Concern # 3

Move the data analysis section at the end of methods section

Author response: Thank you for pointing this out, we have implemented the changes

Reviewer#2, Concern # 4

In tables and results sections, put decimal points instead of commas

Author response: Thank you for pointing this out, modifications have been made.

Reviewer#2, Concern # 5

Table 1. Why were some of the responses divided into for/against and some without division?

Author response: Thank you for this question. Some questions were intentionally specified as agree/disagree in order to categorically assess whether an item should be included or not. Others that specified the direction of change were expanded to include the option of no opinion.

Reviewer#2, Concern # 6

The first sentence of the conclusion is not necessary. Instead, us 'In conclusion…'

Author response: Thank you for pointing this out.

Reviewer#2, Concern # 7

Correct the reference according to the journal requirements

Author response: Thank you for pointing this out. We have made the appropriate changes

Reviewer#2, Concern # 8

The main aim of this study was to analyze the determinants of patients attending an innovative technology-enhanced pulmonary rehabilitation program. Because this program is the most important part of the study, it is worth describing it better. Therefore, other researchers/clinicians will be able to adopt it. I am aware that the authors have indicated another publication in which it is described, but it is not fully described there either. I think it would be best to divide it into 'Conventional Pulmonary Rehabilitation', And 'New Forms of Additions to Pulmonary Rehabilitation'. In these subsections, a more precise description of the exercises performed, the cycloergometer video simulator, relaxation training in virtual reality, cognitive training, and eye-hand coordination in virtual reality is provided.

Author response: Thank you for this comment. We have introduced the division as suggested.

Reviewer#2, Concern # 9

If possible, it is worth supplementing the description of the applied interventions with photos of the therapy.

Author response: Thank you for that suggestion, we added a photo

Reviewer#2, Concern # 10

In addition, I propose to supplement the data analysis if possible. Perhaps supplement patients’ satisfaction, as well as an intergroup comparison. It seems that the authors have data on satisfaction with individual components of rehabilitation, and perhaps it is worth quoting them in the context of new technologies.

Author response: Thank you for this comment. We have supplemented the results section with an analysis of satisfaction with the elements of rehabilitation (Figure 2).

Reviewer#2, Concern # 11

All the cited references were from Poland. This is unacceptable. Please refer to the research results and outline the introduction to international literature.

Author response: Thank you for pointing this out. we have supplemented the references

Reviewer#2, Concern # 12

The limitations of this study should be discussed.

Author response: We added the limitations in the last paragraph of the discussion.

Reviewer#2, Concern # 13

V letter in the title should be written in capital.

Author response: Thank you for pointing this out. we have corrected the notation